# Vibration Responses of a Coaxial Dual-Rotor System with Supporting Misalignment

**Hongxian Zhang** [1] , **Xuejun Li** [2,*], **Dalian Yang** [3] **and Lingli Jiang** [2]

1   School of Mechanical Engineering, Liuzhou Institute of Technology, Liuzhou 454006, China; zhhongxian@163.com
2   School of Mechanical & Electrical Engineering, Foshan University, Foshan 528000, China; linlyjiang@163.com
3   Hunan Provincial Key Laboratory of Health Maintenance for Mechanical Equipment, Hunan University of Science and Technology, Xiangtan 411201, China; hyydl216@163.com
*   Correspondence: hnkjdxlxj@163.com; Tel.: +86-0757-83960006

**Abstract:** In order to improve the thrust-weight ratio, modern aeroengines generally adopt a coaxial dual-rotor system. Factors such as manufacturing errors, assembly errors, bearing wear, and structural deformation can cause misalignment failures in a dual-rotor system. Supporting misalignment is one of the common types of misalignments in a dual-rotor system. To analyze the vibration characteristics of misalignment faults, in this study, we aim to build a finite element model of a dual-rotor system with supporting misalignment. The bearing loads caused by supporting misalignment are calculated using the three-bending moment equation method. Bearing loads are introduced into the dynamic model of the dual-rotor system. The influence of supporting misalignment at different bearings on the dynamic characteristics of the rotor system is investigated based on the supporting misalignment model. Studies have shown that supporting misalignment at different bearings has similar effects on the dynamic characteristics of the dual-rotor system. The proposed supporting misalignment model is more adaptable than the coupling misalignment model. It indicates that the damping of a rolling bearing should be considered in the dynamic analysis of a dual-rotor system although the value of the damping is not large. An experimental analysis is carried out. The simulation results are in good agreement with the experimental results.

**Keywords:** rotor dynamics; dual-rotor system; misalignment; bearing loads; spectral analysis

## 1. Introduction

Misalignment is one of the most common difficulties in the operation of rotating machinery. Misaligned ball bearings in a rigid rotor system have a strong influence on the bearing stiffness coefficients. Angular misalignment of bearings can introduce extra axial force, while parallel misalignment can result in additional moment loads [1]. The parallel misalignment fault of a flexible coupling magnifies the displacement response at 2× amplitude, whereas the angular misalignment response is captured at 2× and 4× amplitudes [2]. Parallel and angular misalignments can both induce second-harmonic generation, but the influence of angular misalignment is more significant than that of parallel misalignment [3]. Cardan shaft misalignment causes an unbalanced force, which is the main load of bolt fatigue fracture of a high-speed train motor [4]. The misalignment of aerostatic journal bearings increases the dynamic stiffness coefficients of a rotor system [5]. Misalignment of a fluid film bearing of finite length increases the value of the critical bearing stability limit [6]. It is obvious that misalignment has a significant influence on the dynamic characteristics of a rotor system. There are many studies on the influence of misalignment on different types of rotor systems. Wang et al. [7] investigated dynamic characteristics of a misaligned rigid rotor system with flexible squirrel cage supports. Ye et al. [8] studied the impact of a gear coupling parallel misalignment on a gear-coupling multi-span rotor system by combining the signal processing method and the finite element

theory. Prainetr [9] presented a method that applied current signal detection and analysis techniques to diagnose an induction motor fault resulting from installation misalignment of a rotor shaft. Kim et al. [10] investigated the heat transfer characteristics of a misalignment under purge flow at second stage vane endwall through heat/mass transfer experiments using the naphthalene sublimation method and CFD simulations. Wu et al. [11,12] investigated the dynamic responses of dynamic spatially misaligned rotors inside a diesel engine multiple unit system. Aggarwal et al. [13] described the incremental inductance method to detect static misalignment of machines. Kumar et al. [14,15] proposed a novel identification algorithm to simultaneously estimate the unbalance, misalignment, and active magnetic bearing stiffness parameters. Xu et al. [16] studied the vibration response characteristics of a generator rotor under misalignment and stator short-circuit coupling fault.

The above studies have achieved significant progress in revealing vibration characteristics of misaligned series-connected rotor systems. However, the dynamics of a dual rotor with misalignment are different from that of series-connected rotor systems which are rotors in an electric motor rotor system [17] and a planetary gearbox rotor system [18]. The different rotors of a series-connected rotor system are connected by a coupling. Each rotor of a series-connected rotor system has the same rotational speed and direction of rotation. On the contrary, a coaxial dual-rotor structure employs an inter-shaft bearing to connect an inner and an outer rotor. These two rotors of a coaxial dual-rotor system can co-rotate or counter-rotate at different speeds. In most cases, the inner rotor is often supported by more than two bearings, making it a redundantly supported system. The special structure of a dual-rotor system coupled with a harsh working temperature environment makes the coaxial dual-rotor system of an aeroengine more prone to misalign. Lu et al. [19] presented a theoretical model of the nonlinear response characteristics of an aeroengine dual-rotor-bearing system with flexible coupling misalignment faults in the inner rotor. Lu et al. [20] applied the proper orthogonal decomposition method to the dimension reduction of a dual-rotor-bearing experimental rig and discussed the frequency behaviors of a dual-rotor-bearing coupling misalignment response. Li et al. [21] described the mechanism and influencing factors of the nonlinear properties of a misaligned inner rotor, in an aeroengine, through Lagrange equations. Wang [22] and Wang [23] et al. developed a multi-bearing rotor model with misalignment to investigate the dynamic response when misalignment parameters were uncertain. Li et al. [24] investigated the quantification of uncertainty effects on the dynamic responses and vibration characteristics of a multi-rotor-bearing system with the fault of angular misalignment. Ren et al. [25] presented a dynamic model of a flexible rotor system with multi-supports that could be used to solve the bearing misalignment problem of a rotor system in an aeroengine. Zhang et al. [26] established a dynamic model of the coupling misalignment of a dual-rotor system according to the Gibbons formula [27] and the Sekhar formula [28]. Yang et al. [29] proposed a dual-rotor misalignment fault quantitative identification method based on DBN and D-S evidence theory improved using the mutual information method. The results showed that the method improved the accuracy of the misalignment fault quantitative identification of the dual rotor. Jin et al. [30] investigated a dual-rotor-bearing system with coupling misalignment and blade-casing rubbing and studied its nonlinear vibration characteristics. Xie et al. [31] investigated the dynamic responses of a dual-rotor system with rubbing–misalignment mixed fault.

Previous studies have presented detailed discussions and analyses of the effect of misalignment on a dual-rotor system of an aeroengine; however, some of the studies considered an inner rotor system instead of the total dual-rotor system. Some of the misalignment models are coupling misalignment models that only explain the impact of the coupling misalignment and do not analyze the effect of the supporting misalignment of an inner rotor which is a multi-bearing rotor system. In response to this shortcoming, in this study, we build a supporting misalignment model of a coaxial dual-rotor system and draw on the three-moment equation. The method has often been used to analyze the effects of misalignment on redundantly supported rotors. For example, Chen [32] and

Yang [33] et al. calculated the bearing loads of a large steam turbine generator set using the three-moment equation. Hori and Uematsu [34] analyzed the influence of misalignment of the support journal bearings on the stability of a multi-rotor system. Hu et al. [35] presented the vibration behavior of statically indeterminate rotor-bearing systems with hydrodynamic journal bearings. Our model based on the three-moment equation can describe misalignment caused by a bearing that is adjacent to the coupling as well as far from the coupling and it is more consistent with the real work condition of an aeroengine. In addition, the experimental results and the simulation results are in good agreement.

## 2. Modeling of a Dual-Rotor System with Supporting Misalignment

### 2.1. Model of a Dual-Rotor System

The structural diagram of a coaxial dual-rotor system is shown in Figure 1. It is a lumped parameter model [26] of an aeroengine (Figure 2). This coaxial dual-rotor system was obtained through the following three steps: First, the solid model of an aeroengine was built by measuring its size. Second, the solid model was imported into a finite element analysis software to obtain the dynamic parameters of the aeroengine. Finally, the dual-rotor system with similar parameters was designed according to the principle of dynamic similarity. The shaft of an inner rotor is solid, and the shaft of an outer rotor is hollow. Regarding the dual-rotor system, the two rotors are connected by an inter-shaft bearing (Bearing 5). The inner race of the inter shaft bearing is mounted on the inner rotor, while the outer race is mounted on the outer rotor. The outer and inner rotors rotate independently, and can rotate at different rotating speeds and in different directions [26]. Disks 1 and 2 represent a three-stage low pressure blade wheel. Disk 3, 4, and 5 represent a low-pressure turbine, a high-pressure compressor, and a high-pressure turbine, respectively. It is inconvenient for the aeroengine prototype to carry out the misalignment test because it is difficult to adjust the degree of misalignment, and the working speed of the aeroengine is also too high for the misalignment test. Therefore, the structural parameters of the lumped parameter model are optimized according to the principle of dynamic similarity. Ignoring the cross stiffness and cross damping coefficients, the bearing stiffness coefficients ($k_{xx}$ and $k_{yy}$) and damping coefficients ($C_{xx}$ and $C_{yy}$) in the horizontal and vertical directions are shown in Table 1. The stiffness of the bearing is much higher than that shown in the table. In order to obtain the bearing stiffness in the table, a special squirrel cage support seat should be designed in the test.

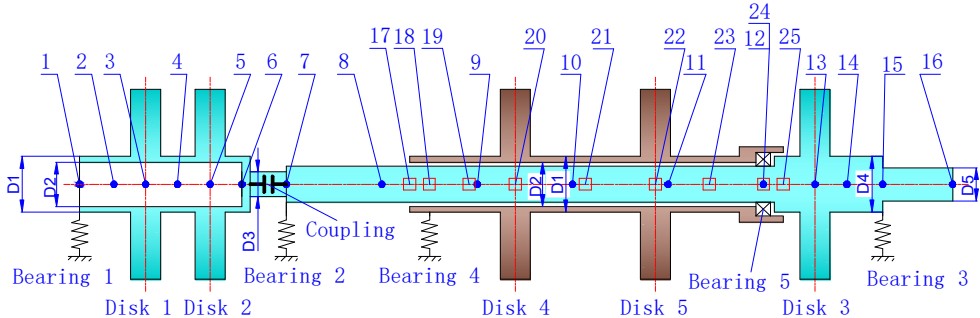

**Figure 1.** Structure diagram of a dual-rotor system with five supports.

**Table 1.** Bearing properties of the rotor system.

| Bearing | $K_{xx}$ (MN/m) | $K_{yy}$ (MN/m) | $C_{xx}$ (Ns/m) | $C_{yy}$ (Ns/m) | Specification |
|---|---|---|---|---|---|
| Bearing 1 | 2.21 | 2.651 | 200 | 200 | NU1013 |
| Bearing 2 | 14.5 | 17.4 | 200 | 200 | 7013AC |
| Bearing 3 | 2.21 | 2.651 | 200 | 200 | NU1013 |
| Bearing 4 | 9.29 | 11.148 | 200 | 200 | 7013AC |
| Bearing 5 | 25.1 | 25.1 | 200 | 200 | NU1013 |

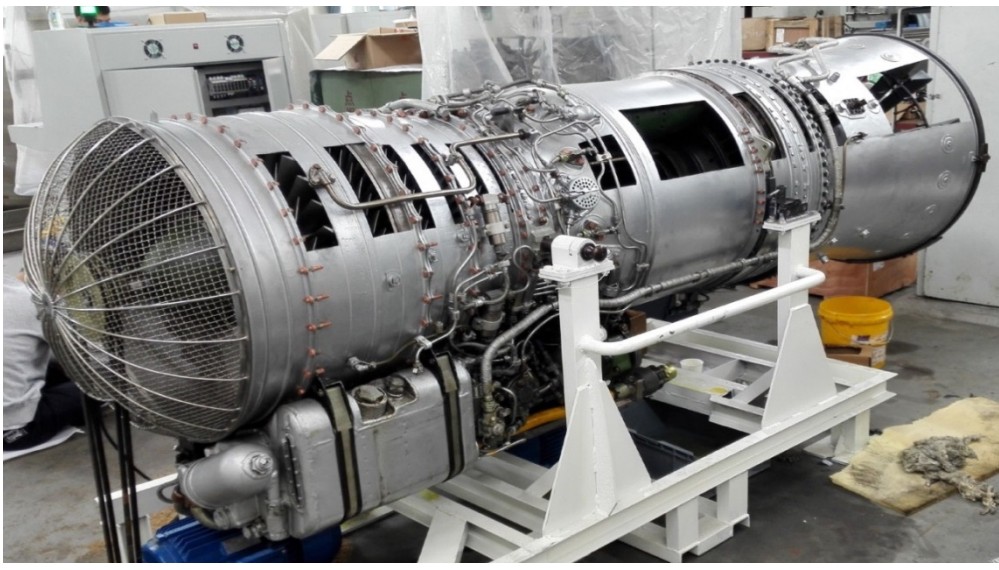

**Figure 2.** Prototype of an aeroengine.

The sections of the inner and outer rotor shafts are all circular. The section diameters are expressed as D1, D2, D3, D4, and D5, respectively. The diameters of the shafts are shown in Table 2.

**Table 2.** Diameters of the shafts.

| Parameters | D1 (m) | D2 (m) | D3 (m) | D4 (m) | D5 (m) |
|---|---|---|---|---|---|
| Values | 0.07 | 0.056 | 0.035 | 0.07 | 0.042 |

The five discs have the same size. The diameter, thickness, elastic modulus, and density of each disk are shown in Table 3.

**Table 3.** Disk properties for the rotor system.

| Parameters | Diameter (m) | Thickness (m) | Elastic Modulus (Mpa) | Density (kg/m$^3$) |
|---|---|---|---|---|
| Values | 0.24 | 0.038 | $2.06 \times 10^5$ | 7850 |

The dual-rotor system is modeled by the finite element method [36]. The inner rotor is modeled with 15 elements and 16 nodes. The outer rotor is modeled with 8 elements and 9 nodes. Thus, the model has 23 elements and 25 nodes in total. The nodes of the inner rotor are denoted by blue dots. The nodes of the outer rotor are denoted by red squares, which are shown in Figure 1. The node locations are listed in Table 4.

Assembling all the damping matrices, stiffness matrices, and mass matrices of elements, the general dynamic equations of the dual-rotor system can be written as [36]:

$$M\ddot{q} + (C + \Omega_1 G_1 - \Omega_2 G_2)\dot{q} + Kq = F_g + F_u + F_b \tag{1}$$

where $M$, $C$, and $K$ are the mass, damping, and stiffness matrices of the dual-rotor system, respectively; $q$ is the displacement vector; $\Omega_1$ and $\Omega_2$ are the rotation speed of the inner rotor and the outer rotor, respectively; $G_1$ and $G_2$ are the gyroscopic matrixes of the inner and the outer rotor, respectively; $F_g$ is the gravity vector; $F_u$ is the vectors of the unbalance excitation of the rotor system; $F_b$ is the bearing load vector caused by misalignment of bearings, which will be derived in Section 2.2.

**Table 4.** Node location.

| Node | Position (m) | Remark | Node | Position (m) | Remark |
|------|--------------|--------|------|--------------|--------|
| 1 | 0.000 | Bearing 1 | 14 | 0.965 | |
| 2 | 0.043 | | 15 | 1.010 | Bearing 3 |
| 3 | 0.083 | Disk 1 | 16 | 1.098 | |
| 4 | 0.123 | | 17 | 0.415 | |
| 5 | 0.164 | Disk 2 | 18 | 0.440 | Bearing 4 |
| 6 | 0.204 | | 19 | 0.490 | |
| 7 | 0.260 | Bearing 2 | 20 | 0.548 | Disk 4 |
| 8 | 0.380 | | 21 | 0.636 | |
| 9 | 0.500 | | 22 | 0.724 | Disk 5 |
| 10 | 0.620 | | 23 | 0.792 | |
| 11 | 0.740 | | 24 | 0.860 | Bearing 5 |
| 12 | 0.860 | Bearing 5 | 25 | 0.885 | |
| 13 | 0.925 | Disk 3 | | | |

*2.2. Bearing Load Caused by Misalignment*

The inner rotor of the coaxial dual-rotor system is supported by three bearings, and therefore it is a redundantly supported rotor. The support scheme of the inner rotor is shown in Figure 3. The coupling of the rotor system is a special coupling for an aeroengine which has stronger stiffness than a common gear coupling [21]. This structure causes the bearing load to be redistributed when any bearing support is misaligned. Ignoring the change in the section size of the shaft, the inner rotor is regarded as a uniform beam. The three-moment equation method is employed to calculate the bearing load caused by the supporting bearing misalignment [32].

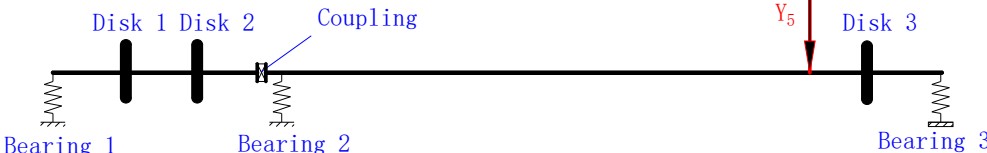

**Figure 3.** The support scheme of the inner rotor.

**(1)    Bearing Load of the Outer Rotor**

The dual rotor is decomposed into an inner beam and an outer beam according to its force analysis. The force analysis of the inner beam is shown in Figure 4. The bending moment balance equation can be obtained as follows:

$$\begin{cases} Y_5 L_{45} - m_5 g a_{42} - m_4 g a_{41} - \frac{1}{2}(L_{45})^2 q_{45} = 0 \\ Y_4 L_{45} - m_4 g(L_{45} - a_{41}) - m_5 g(L_{45} - a_{42}) - \frac{1}{2}(L_{45})^2 q_{45} = 0 \end{cases} \tag{2}$$

where $Y_4$ and $Y_5$ are the supporting loads of Bearings 4 and 5, respectively; $L_{45}$ is the distance between the two bearings; $m_4$ and $m_5$ represent the mass of Disks 4 and 5, respectively; $a_{41}$ and $a_{42}$ represent the distance from Disks 4 and 5 to Bearing 4, respectively; $q_{45}$ is the uniformly distributed mass of the outer rotor; g is the acceleration of gravity.

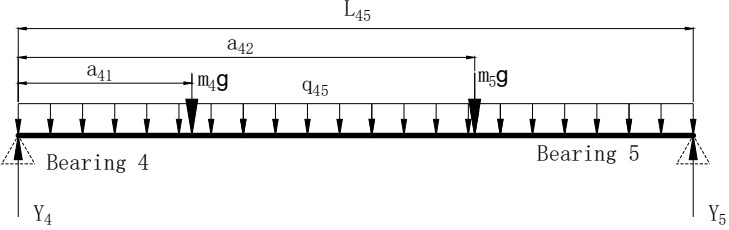

**Figure 4.** Applied forces of the outer rotor.

**(2)    Bearing Load of the Inner Rotor**

The force analysis of the inner rotor is shown in Figure 5. The inner rotor is a redundantly supported beam. In order to obtain the bearing load, the fixed constraint of Bearing 2 is replaced by bearing load $Y_2$ and a pair of bending moments $M_2$, and then the redundantly supported beam is decomposed into two simply supported beams. Finally, bearing loads of the inner rotor can be calculated by the deformation compatibility principle and the moment balance equation.

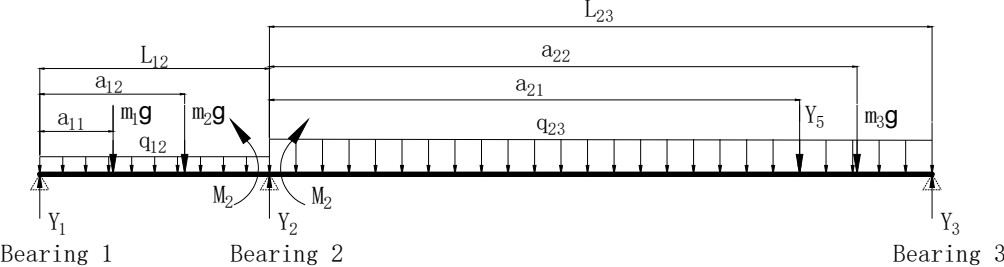

**Figure 5.** Applied forces of the inner rotor.

According to the deformation compatibility principle, the deflection angles caused by both various loads and bearing misalignment are equal on the left and right end faces of Bearing 2. It is stipulated that the counterclockwise deflection angles are positive and the clockwise deflection angles are negative. Thus, the deflection angle equations can be obtained as follows:

$$\theta_2^R = \frac{-L_{23}M_2}{3EI_{23}} + \frac{Y_5 a_{21}(L_{23} - a_{21})(2L_{23} - a_{21})}{6L_{23}EI_{23}} - \frac{m_3 g a_{22}(L_{23} - a_{22})(2L_{23} - a_{22})}{6L_{23}EI_{23}} - \frac{q_{23}(L_{23})^3}{24EI_{23}} - \frac{\Delta_3}{L_{23}} + \frac{\Delta_2}{L_{23}} \quad (3)$$

$$\theta_2^L = \frac{L_{12}M_2}{3EI_{12}} + \frac{m_1 g a_{11}(L_{12} - a_{11})(L_{12} + a_{11})}{6L_{12}EI_{12}} + \frac{m_2 g a_{12}(L_{12} - a_{12})(L_{12} + a_{12})}{6L_{12}EI_{12}} + \frac{q_{12}(L_{12})^3}{24EI_{12}} - \frac{\Delta_2}{L_{12}} + \frac{\Delta_1}{L_{12}} \quad (4)$$

where $\theta_2^R$ and $\theta_2^L$ are the deflection angles of the right and left ends of Bearing 2, respectively; $L_{12}$ and $L_{23}$ are the bearing spacing; $M_2$ is the bending moment caused by the over-constraint of Bearing 2; $E$ is the elastic modulus of the shaft; $I_{12}$ and $I_{23}$ are the moment of inertia; $a_{11}$, $a_{12}$, $a_{21}$, and $a_{22}$ represent the distances from the different loads to Bearings 1 and 2, respectively; $\Delta_1$, $\Delta_2$, and $\Delta_3$ represent the misalignment of the three bearings, respectively. It is specified that the misalignment is negative if the bearing is padded up.

Set $\theta_2^R$ equal to $\theta_2^L$, and obtain the bending moment M2. Then, take each simply supported beam of the inner rotor as the research object, and then the moment balance equations are obtained as follows:

$$\begin{cases} Y_2^L L_{12} + M_2 - m_1 g a_{11} - m_2 g a_{12} - \frac{1}{2}(L_{12})^2 q_{12} = 0 \\ -Y_1 L_{12} + M_2 + m_1 g(L_{12} - a_{11}) + m_2 g(L_{12} - a_{12}) + \frac{1}{2}(L_{12})^2 q_{12} = 0 \end{cases} \quad (5)$$

$$\begin{cases} -Y_2^R L_{23} - M_2 - Y_5(L_{23} - a_{21}) + m_3 g(L_{23} - a_{22}) + \frac{1}{2}(L_{23})^2 q_{23} = 0 \\ Y_3 L_{23} - M_2 + Y_5 a_{21} - m_3 g a_{22} - \frac{1}{2}(L_{23})^2 q_{23} = 0 \end{cases} \quad (6)$$

where $m_1$, $m_2$, and $m_3$ represent the masses of Disks 1, 2, and 3, respectively; $Y_2^L$ and $Y_2^R$ are the shear forces on the left and right faces of Bearing 2, respectively; $q_{12}$ is the uniform distributed mass of the inner rotor.

**(3)    Dynamic Bearing Load Caused by Misalignment**

The bearing loads from the above method are static force. However, for a rotating shaft, these static forces become periodic dynamic forces which cause vibrations. According

to the Fourier series expansion method, these periodic dynamic forces can be expanded into a series of harmonic forces [37,38]. The dynamic bearing loads are treated as excitations at the bearing nodes of the finite element model. Only the $1\times$, $2\times$, $3\times$, and $4\times$ harmonic components are considered in this study. The dynamic bearing loads are given as follows:

$$\boldsymbol{F}_b = \begin{bmatrix} \boldsymbol{F}_b^1 & \boldsymbol{F}_b^2 & \cdots & \boldsymbol{F}_b^{25} \end{bmatrix}^T \tag{7}$$

$$\boldsymbol{F}_b^j = \begin{bmatrix} 0 & 0 & 0 & 0 \end{bmatrix}, (j \neq 1, 7, 15.) \tag{8}$$

$$\boldsymbol{F}_b^j = \begin{bmatrix} 0 \\ Y_i \left( \cos(\Omega_1 t) + \left(\tfrac{1}{2}\right)^2 \cos(2\Omega_1 t) + \left(\tfrac{1}{3}\right)^2 \cos(3\Omega_1 t) + \left(\tfrac{1}{4}\right)^2 \cos(4\Omega_1 t) \right) \\ 0 \\ 0 \end{bmatrix}^T \tag{9}$$

where $\boldsymbol{F}_b^j$ is the dynamic bearing load of the misalignment at node $j$, where $i = 1, j = 1$; $i = 2$, $j = 7$; $i = 3, j = 15$.

### 3. Simulation and Discussions

Assume that the outer rotor counter rotates with respect to the inner rotor at a speed of 1.6 times. The inner rotor rotates at 660 rpm (11 Hz). The outer rotor rotates at 800 rpm (17.6 Hz). Disks 2, 3, an 5 have unbalance forces of magnitude 0.0003, 0.0002, 0.0003 kg/m$^3$, respectively. The spectral characteristics of the dual-rotor system are studied when Bearings 1 and 3 are assumed to be 0.2 mm ($\Delta_1 = 0.2$ mm and $\Delta_3 = 0.2$ mm), respectively. The influence of bearing damping and Fourier expansion coefficients on dynamic characteristics of the dual-rotor system is analyzed.

### 3.1. Effects of Supporting Misalignment at Bearing 1

The vibration displacement and orbits of Bearing 1 are shown in Figure 6, when Bearing 1 is padded to 0.2 mm in the vertical direction and it can be seen that misalignment makes the orbit of Bearing 1 (node 1) present the shape of the Arabic number "8". The time-domain waveform diagram in the vertical direction (misalignment direction) is more complicated than that of the horizontal direction.

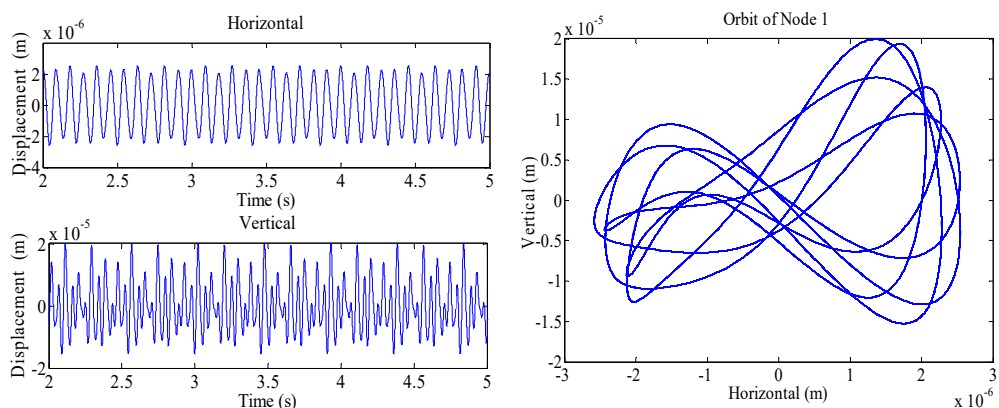

**Figure 6.** Vibration displacement and the orbit of Bearing 1 (misalignment at Bearing 1).

The spectrum of Bearing 1 is shown in Figure 7 when Bearing 1 is padded up. The direction of the dynamic bearing load caused by misalignment does not rotate the rotor. Therefore, the additional force of misalignment mainly affects the vibration in the direction of misalignment. The vibration spectrum in the direction without misalignment changes little. The horizontal spectrum in Figure 7 shows only the frequency of the first-order vibration of the inner rotor (represented as $1\times$) and the first-order vibration of the outer rotor (represented as $1.6\times$) in the horizontal direction. However, misalignment affects

the spectrum of the vertical direction significantly because the intention misalignment is set in the vertical direction. In addition to the frequencies of $1\times$ and $1.6\times$, there are second-, third-, and fourth-order vibrations of the inner rotor, which are represented as $2\times$, $3\times$, and $4\times$, respectively. Among them, $1\times$, $1.6\times$, and $2\times$ are the main vibration frequencies. The above analysis reveals that the influence of misalignment on the rotor system is mainly reflected in the misalignment direction. Therefore, for the analysis of the other three bearings, we only list the frequency spectrum and the orbit in the vertical direction (misalignment direction).

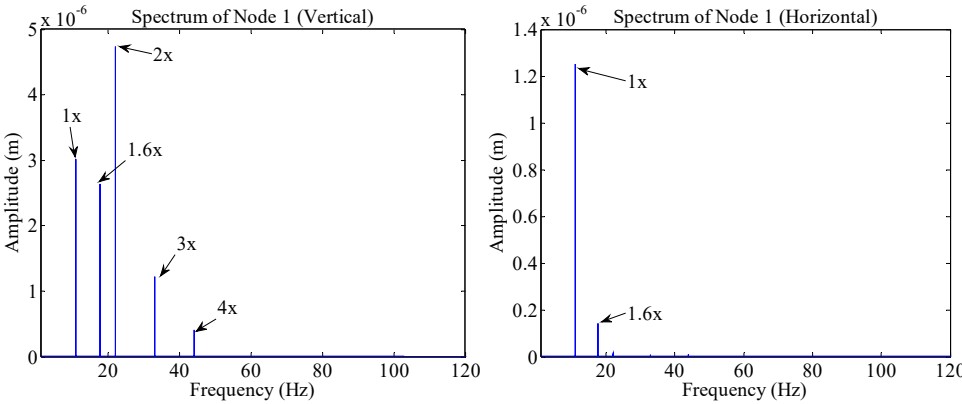

**Figure 7.** The spectrum of Bearing 1 (misalignment at Bearing 1).

The frequency spectra and orbits of Bearings 3 and 4 are shown in Figures 8 and 9, respectively, and it can be seen that the frequency components and the shape of the orbits at the two bearings are similar to those of Bearing 1. The inter-shaft Bearing 5 attenuates the vibration transmission between the inner and the outer rotor, which causes the frequency $2\times$ of Bearing 4 on the outer rotor to be weaker than that of the three bearings of the inner rotor. Despite the vibration attenuating effect, the misalignment of the inner rotor Bearing 1 will also affect the outer rotor. The vibration effects of Bearings 2 and 1 are similar. Due to the limitation of the experimental conditions, the subsequent tests do not measure the vibration of Bearing 2 separately. Therefore, the frequency spectrum and the orbit of Bearing 2 are not listed separately.

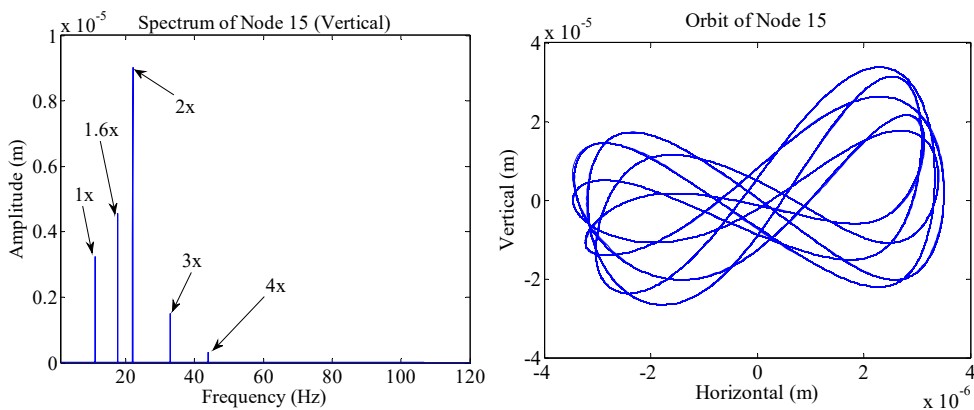

**Figure 8.** The spectrum and the orbit of Bearing 3 (misalignment at Bearing 1).

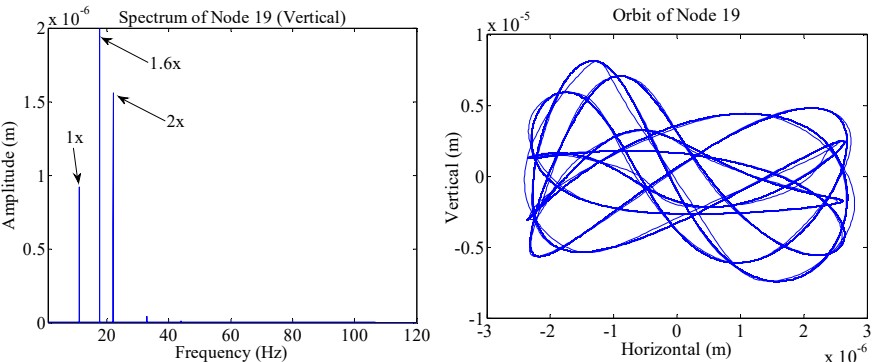

**Figure 9.** The spectrum and the orbit of Bearing 4 (misalignment at Bearing 1).

### 3.2. Effects of Supporting Misalignment at Bearing 3

In the coupling misalignment model [26], it shows that the supporting misalignment of Bearing 3 has no effect on the rotor system because the bearing elevation of Bearing 3 causes no misalignment in the coupling. This is inconsistent with the actual test results, which are discussed in Section 4. Therefore, the coupling model can only explain the impact of the supporting misalignment of Bearings 1 and 2 on the dynamic characteristics of the rotor system, while the model of bearing supporting misalignment can fill up the defect.

Supposing the supporting misalignment occurs at Bearing 3, the orbits and vertical spectra of Bearings 1, 3, and 4 are shown in Figures 10–12, respectively. The figures show that the impact of the supporting misalignment of Bearing 3 and that of Bearing 1 on the frequency components of the rotor system are basically the same. The frequency spectrum is also dominated by $1\times$, $1.6\times$, and $2\times$. Although $3\times$ and $4\times$ are also aroused, these two frequencies are relatively weak. The differences of misalignment effects of Bearings 1 and 3 can be drawn as follows:

(1) The amplitude of $2\times$ caused by the misalignment of Bearing 3 is smaller than that of Bearing 1. The vibration caused by the misalignment of Bearing 1 is more severe when Bearings 1 and 3 have the same degree of misalignment. In other words, the coaxial dual-rotor system is more sensitive to misalignment at Bearing 1, which may be due to the smaller bearing load generated by Bearing 3 under the same misalignment.

(2) The orbit is in the shape of a "reticulate spindle" when Bearing 3 is misaligned. However, the 8-shaped orbit is often regarded as a typical feature of misalignment, which is similar to the orbit when Bearing 1 is misaligned. This may be due to the smaller $2\times$ amplitude.

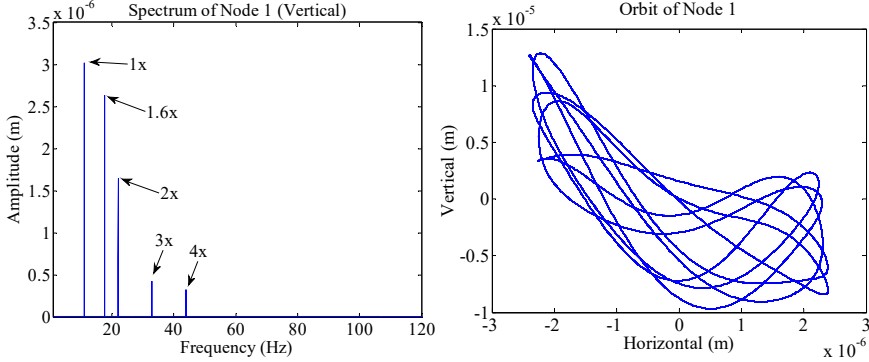

**Figure 10.** The spectrum and the orbit of Bearing 1 (misalignment at Bearing 3).

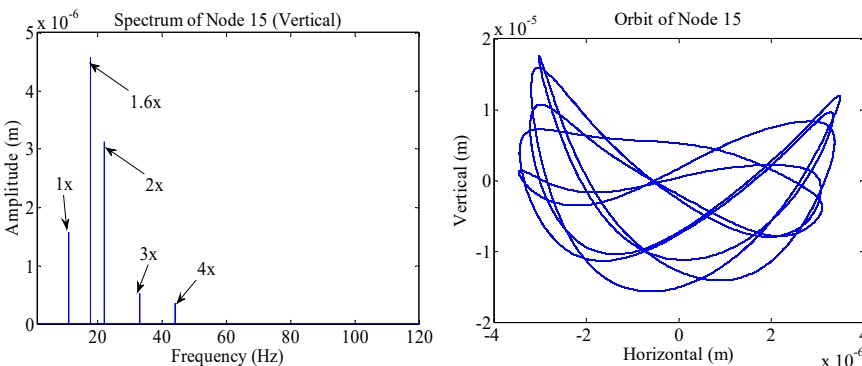

**Figure 11.** The spectrum and the orbit of Bearing 3 (misalignment at Bearing 3).

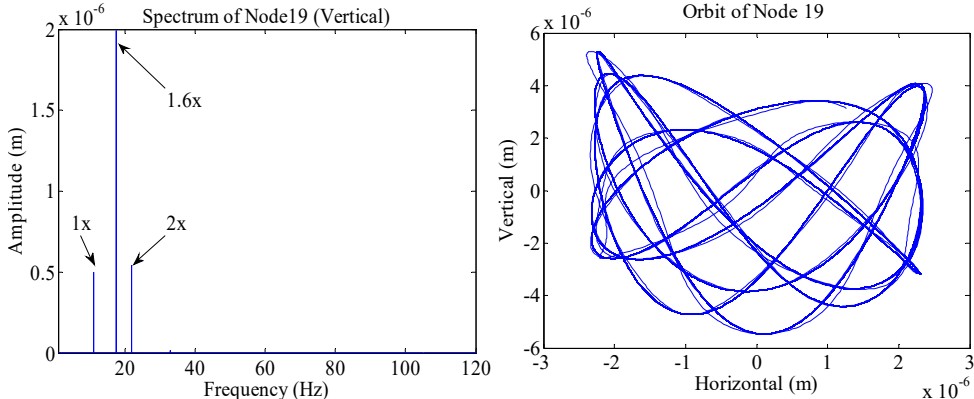

**Figure 12.** The spectrum and the orbit of Bearing 4 (misalignment at Bearing 3).

### *3.3. Effects of Bearing Damping*

In previous studies [26], it has been found that coupling misalignment can excite several backward and forward natural frequencies of a dual-rotor system when the bearing damping is zero. In the present analysis, the bearing damping is set to 200 Ns/m, and the backward and forward natural frequencies are not excited, which are described in Sections 3.1 and 3.2. In order to verify whether this problem is caused by the bearing damping factor, the following comparative research is performed. All parameters remain unchanged except that Bearing 1 is elevated for an intentional misalignment. The damping of all bearings is changed from 200 to 0 Ns/m. The frequency spectra of Bearings 1, 3, and 4 are shown in Figures 13 and 14. By comparing the two figures with Figures 7–9, the following can be stated:

(1) From the observations of the frequency spectra, it can be seen that the bearing damping suppresses the natural frequency of forward and back precession. The spectra of Bearings 1 and 3 show the notable first-order backward natural frequency (BW1) and the second-order forward natural frequency (FW2) when the damping is zero. Meanwhile, the spectrum of Bearing 4 shows the frequency components of BW1, FW1, and BW2. These natural frequencies all disappear when the damping is increased to 200 Ns/m, only the frequency components such as $1\times$, $1.6\times$, and $2\times$ are left. The bearing damping does not change the relative magnitude of $1\times$, $1.6\times$, and $2\times$, but only suppresses the natural frequencies.

(2) Regarding the orbit, when the damping is zero, the orbit becomes a complicated "reticulated structure" shape instead of a simple 8-shaped orbit. In the research, it is found that the shape of the orbit is affected by many factors, such as the damping of the bearing, the rotational speeds of the inner and the outer rotor, the unbalanced mass of the inner and the outer rotor, the amount of misalignment, and the position of other misalignments. Therefore, it is inaccurate to judge the misalignment error only by the shape of the orbit,

especially for a dual-rotor system. It will be interesting to explore the changing law of the orbit of the dual-rotor system in the future.

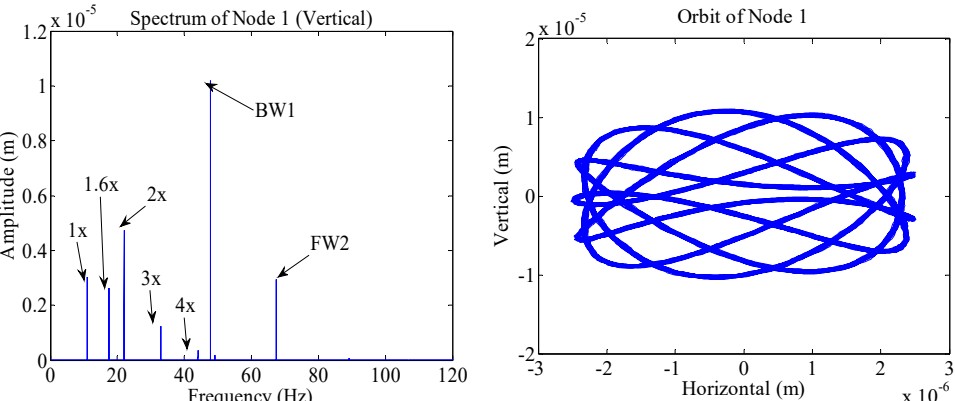

**Figure 13.** The spectrum and the orbit of Bearing 1 (without bearing damping).

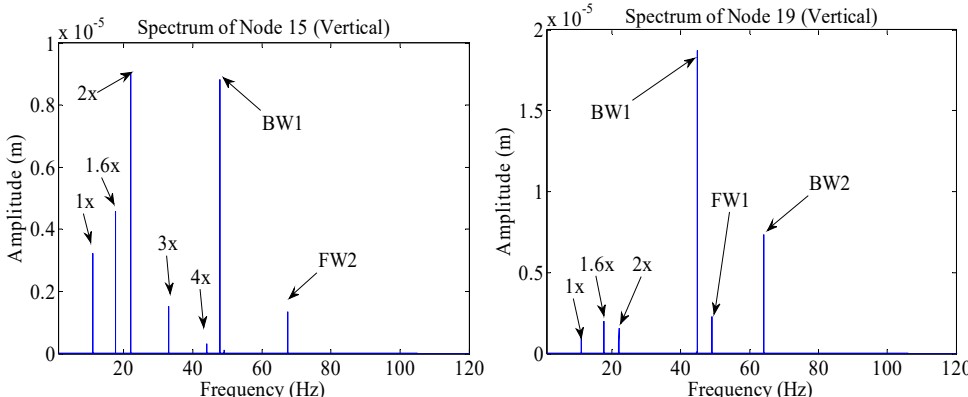

**Figure 14.** The spectrum of Bearing 3 and Bearing 4 (without bearing damping).

### 3.4. Effects of the Coefficient of the Harmonic Component

The bearing load is a static force to the non-rotating observer. However, it acts as a periodic load on the rotating rotor. According to the Fourier analysis, any periodic function or periodic signal can be decomposed into the sum of a (possibly infinite) set of simple oscillating functions, namely sines and cosines. Ma [37], Zhao [38], and Prabhakar [39] et al. decomposed the periodic misalignment force into the sum of harmonic components $1\times$, $2\times$, $3\times$, and $4\times$. The four harmonic components have equal weight coefficients. In Section 2.2, the weight coefficients of these four harmonic frequencies are set to 1, 1/4, 1/9, and 1/16, respectively, because the latter is in line with the actual situation of a dual-rotor system. The weight coefficients of the four harmonic components are set to be the same in order to compare the difference between the two cases. The dynamic bearing load caused by misalignment can be expressed as follows:

$$
\boldsymbol{F}_b^j = \begin{bmatrix} 0 \\ Y_i(\cos(\Omega_1 t) + \cos(2\Omega_1 t) + \cos(3\Omega_1 t) + \cos(4\Omega_1 t)) \\ 0 \\ 0 \end{bmatrix}^T
\tag{10}
$$

The orbit and frequency spectrum of each bearing are shown in Figures 15 and 16 when the weight coefficients of the four harmonic components are equal. Comparing the two figures with Figures 7–9, it is shown that the amplitudes of $2\times$, $3\times$, and $4\times$ at the three bearings have all increased significantly. The maximum vibration frequencies of

Bearing 1 and Bearing 3 of the inner rotor change from $1\times$, $1.6\times$, and $2\times$ to $2\times$, $3\times$, and $4\times$. The shape of the orbit also becomes more complicated due to the large change in the vibration frequency component. The experimental results in Section 4 indicate that the non-uniform weight coefficient model (Equation (9)) is more reasonable than the equal weight coefficients model (Equation (10)).

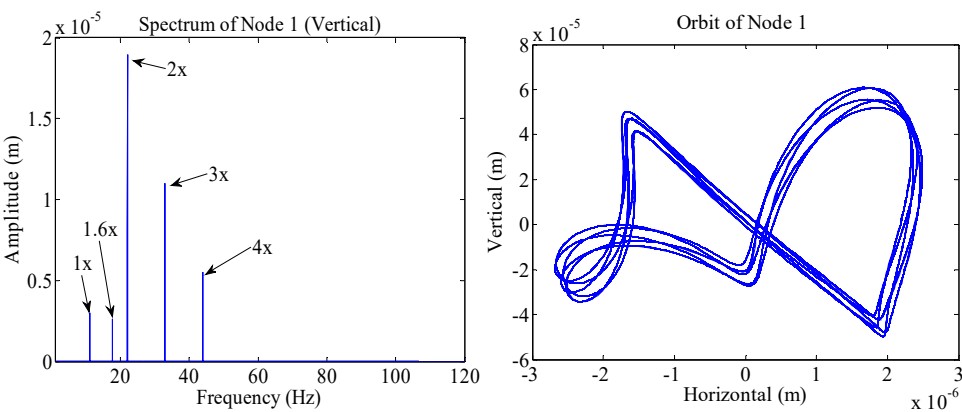

**Figure 15.** The spectrum and the orbit of Bearing 1 (equal weight coefficients).

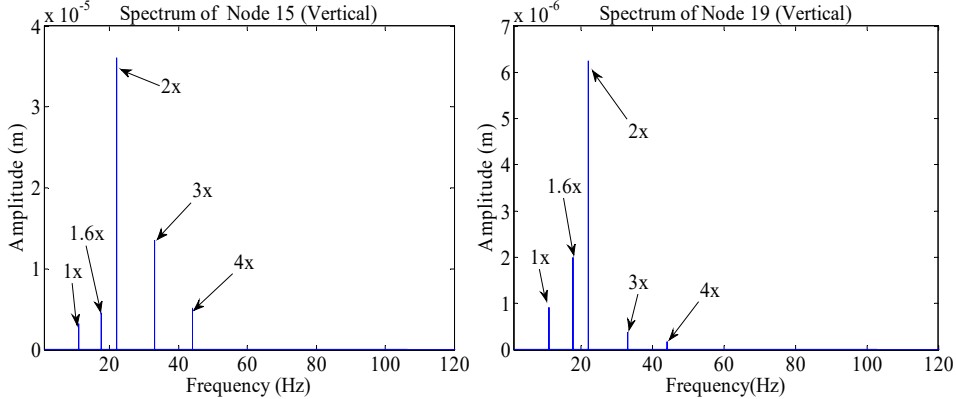

**Figure 16.** The spectrum of Bearing 3 and Bearing 4 (equal weight coefficients).

## 4. Experimental Study of Bearing Supporting Misalignment

A dual-rotor test rig is designed and manufactured to verify the simulation results, which is shown in Figure 17. Its structural parameters are the same as those for the test bench described in Figure 1. The outer rotor motor drives the outer rotor by a cog belt. The inner rotor motor drives the inner rotor by a coupling. The speed ratio between the inner and outer rotors can be adjusted arbitrarily. It is assumed that the speed of the outer rotor is 1.6 times that of the inner rotor. Experiments were performed on the test rig under the intentional misalignment of Bearing 1 and Bearing 3, respectively. Bearing 1 and Bearing 3 are separately tested with a pad height of 0.2 mm.

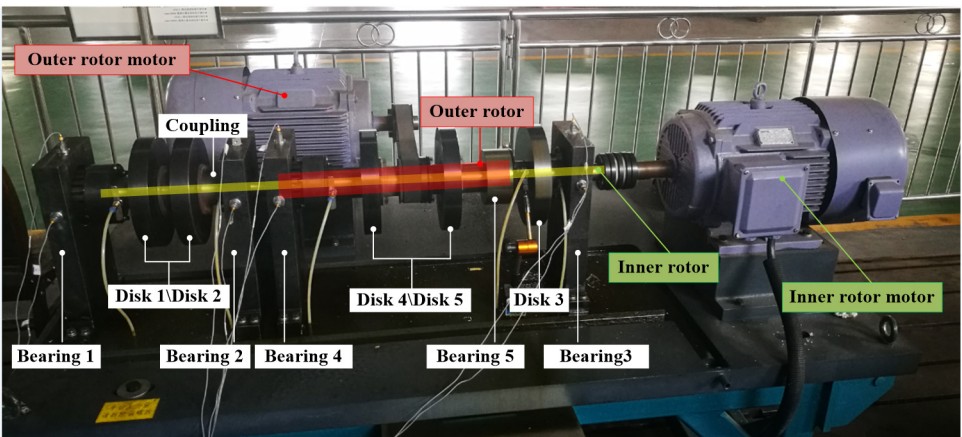

**Figure 17.** Experimental setup of a dual-rotor system.

Vibration data are acquired from the four bearing pedestals using a B&K data collector. Both horizontal and vertical vibrations of the bearing pedestals are measured by accelerometers. The acceleration signal collected by the sensor is firstly converted into a displacement signal by double integration. The FFT algorithm is applied on the displacement signal to obtain the spectrum. A high-pass filter is adapted to filter low-frequency noise signals below 8 Hz, because the low-frequency noise will be generated in this process.

### 4.1. Supporting Misalignment at Bearing 1

The orbits and frequency spectra of the rotor system are shown in Figures 18–21 when Bearing 1 is intentionally elevated. Observations of these figures indicates the following:

(1) Both the oscillogram and the spectrogram show that the vibration in the vertical direction (misalignment direction) is more complicated than in the horizontal direction. The frequency components in the vertical direction are dominated by $1\times$, $1.6\times$, and $2\times$. The orbits at the three bearings are 8-shaped. These are consistent with the simulation results.

(2) Although there are $3\times$ and $4\times$ in the spectrograms, the amplitudes are relatively small. This is inconsistent with the simulation results of the equal weight coefficients model. It confirms that the non-uniform weight coefficient model proposed in this study is more in line with the actual situation.

(3) All bearings have no obvious natural frequency. This suggests that although the damping of the rolling bearing is small, it still cannot be ignored in the simulation process.

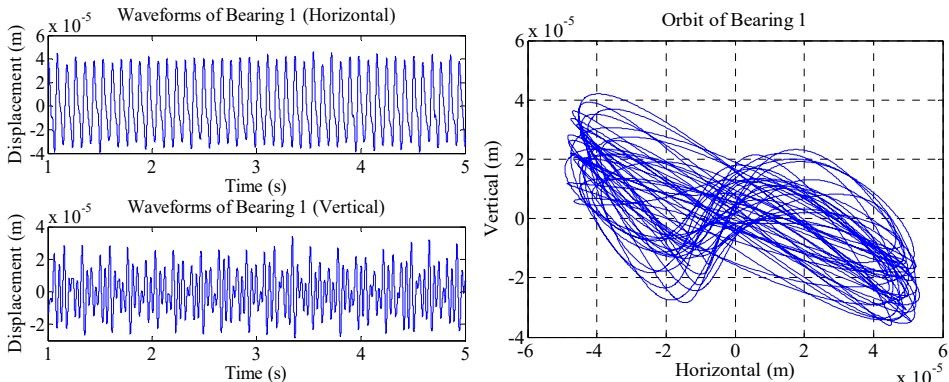

**Figure 18.** The vibration displacement and orbit of Bearing 1.

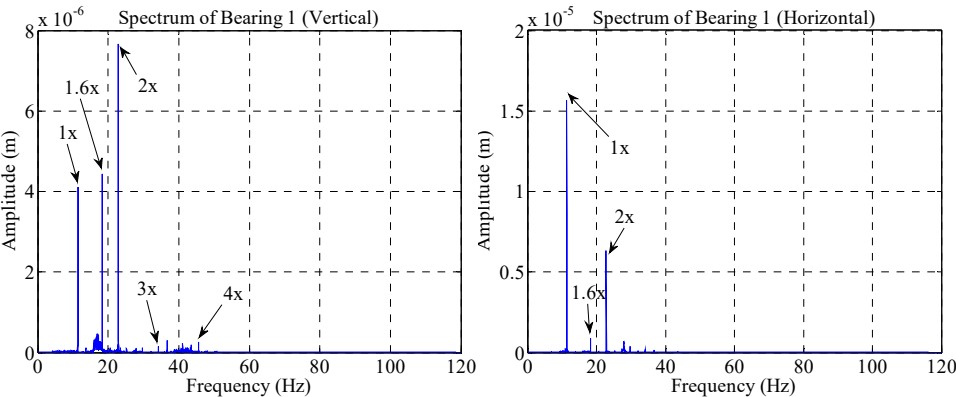

**Figure 19.** The spectrum of horizontal and vertical direction of Bearing 1.

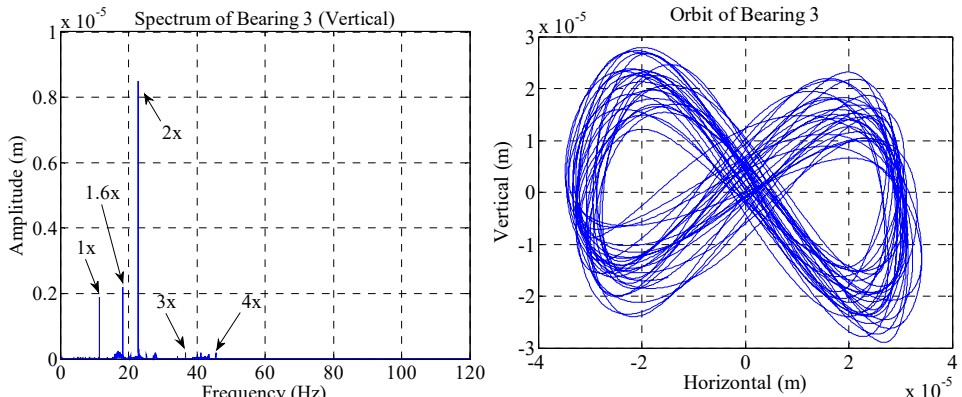

**Figure 20.** The spectrum and the orbit of Bearing 3.

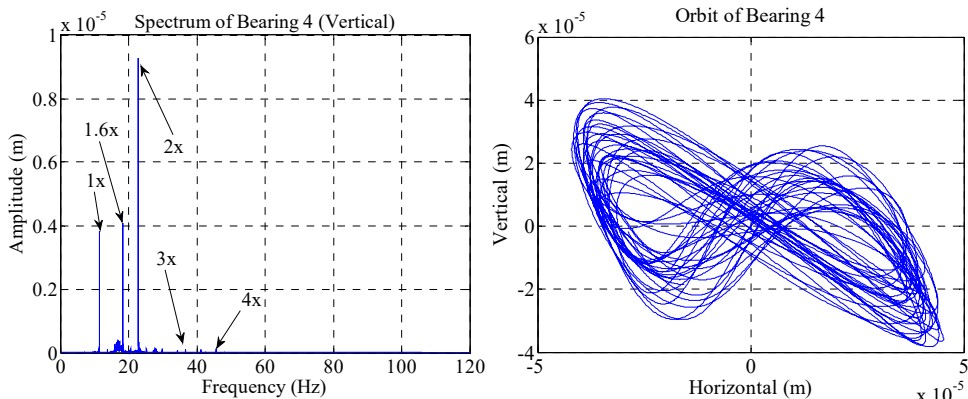

**Figure 21.** The spectrum and the orbit of Bearing 4.

*4.2. Supporting Misalignment at Bearing 3*

When Bearing 1 is intentionally elevated, the orbit and frequency spectra of the rotor system are shown in Figures 22–24. The figures indicate that the spectral characteristics of the misalignment at Bearings 3 and 1 are similar. It confirms that the supporting misalignment of Bearing 3 will also affect the rotor system. However, the dynamic behavior caused by the misalignment of the two bearings is also somewhat different. One difference is that the orbit is no longer 8-shaped, but a "banana shape". Another difference is that the overall amplitude caused by Bearing 3 misalignment is smaller than that of Bearing 1 misalignment. The amplitude of 2× also decreases slightly. All in all, these experimental results are basically consistent with the simulation results, which identify the correctness of the supporting misalignment model proposed in this study.

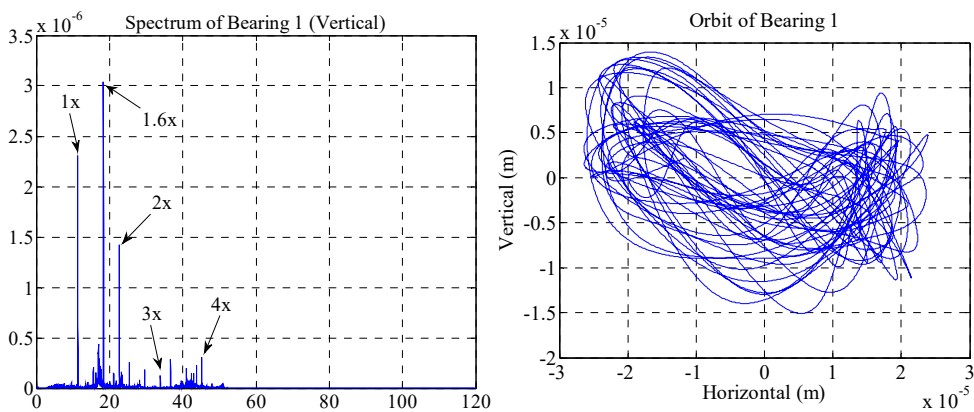

**Figure 22.** The spectrum and the orbit of Bearing 1.

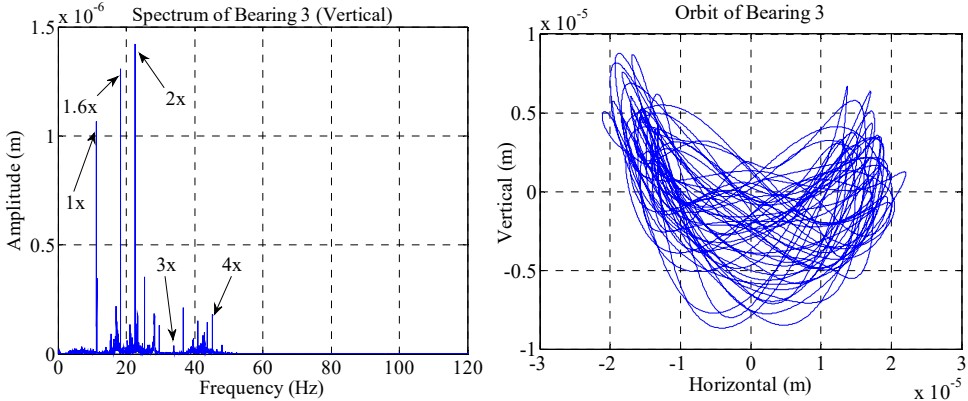

**Figure 23.** The spectrum and the orbit of Bearing 3.

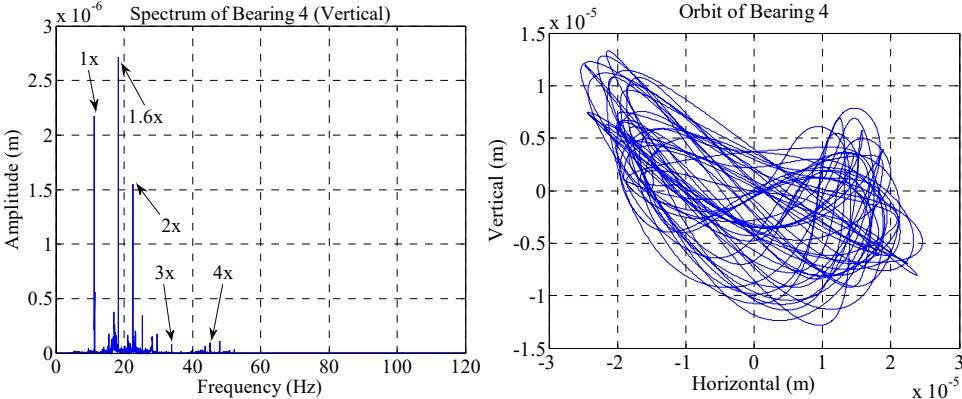

**Figure 24.** The spectrum and the orbit of Bearing 4.

However, in addition to the frequency components predicted by simulation such as $1\times$, $1.6\times$, and $2\times$, there are other frequency components in the measured results. These frequency components may be caused by nonlinear factors of the system, such as nonlinear vibration caused by bolt looseness and bearing clearance.

## 5. Conclusions

In this study, we build the finite element model of a coaxial dual-rotor system with supporting misalignment. The bearing load caused by bearing supporting misalignment is calculated according to the three-moment equation method. Then, the influence of supporting misalignment on the spectral characteristics and orbits of the dual-rotor system

are analyzed. Finally, an experimental analysis is carried out. The simulation results are consistent with the experimental results. Some conclusions drawn from the study can be summarized as follows:

(1) The proposed supporting misalignment model is more applicable than the coupling misalignment model. The supporting misalignment model can describe the effect of misalignment of any one of the three bearings of Bearings 1, 2, and 3 on the rotor system. However, the coupling misalignment model can only describe the influence of misalignment caused by Bearings 1 and 2, close to the coupling.

(2) Supporting misalignments of Bearings 1 and 3 have similar effects on a dual-rotor system. The main vibration frequencies are $1\times$, $1.6\times$, and $2\times$, with the weaker $3\times$ and $4\times$. However, the vibration caused by the misalignment of Bearing 3 is weaker than that of Bearing 1.

(3) The damping of the rolling bearing should be considered in the dynamic simulation process, although the damping of the rolling is very small.

**Author Contributions:** Conceptualization, methodology, software, writing—original draft preparation, H.Z.; validation, formal analysis, D.Y.; investigation, resources, X.L.; data curation, D.Y. and H.Z.; writing—review and editing, H.Z.; visualization, L.J.; supervision, L.J.; project administration, funding acquisition, X.L. All authors have read and agreed to the published version of the manuscript.

**Funding:** This work is supported by the Natural Science Foundation of China through the grants (51875196 and 12072076).

**Institutional Review Board Statement:** Not applicable.

**Informed Consent Statement:** Not applicable.

**Data Availability Statement:** Not applicable.

**Acknowledgments:** The authors would like to thank Qing Xiao for his valuable suggestion, the graduate students Fanyu Zhang and Jingjing Miao for their help in experiments, and Linlin Song for help in translation work.

**Conflicts of Interest:** The authors declare that there are no conflicts of interest regarding the publication of this paper.

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
