# Peer review of "Vibration Responses of a Coaxial Dual-Rotor System with Supporting Misalignment"

_applsci, doi:10.3390/app112311219_

Round 1

Reviewer 1 Report

A. The authors presented the results of scientific investigation on a specific dual-rotor system, a model of aero-engine with some additional effects (misalignment and loads). The work is devoted to the analysis of steady state in the systems, where the misalignment is taken into account. The dynamics of the system (i.e. transient states) are not discussed. Therefore the Fourier transform is applied and spectral analysis is the basic form of representation of state of the system.

B. The research was carried out on the basis of the selected system, however, the presented results and the text of the paper contain some elements of generalization. In this way, the presented results can be useful for other researchers and can be implemented to other systems.

C. I assess correctly the methodology of the research work. The authors presented the analytical model of the system (differentia equations) and the construction of the numerical model based on the finite element method. The FEM scheme is the primary tool used by the authors. The results of numerical simulations were verified using results of measurements on a laboratory stand.

D. The general scheme of the paper is correct (discussion of the other works, presentation of the problem, solution, etc.). The implemented notation is correct and consistent. The language of the paper is correct. The article requires some corrections. Detailed remarks regarding errors in notation, form of text, and wording are marked in the attached file.

E. Below I present some critical comments on the results of the work.

  1. The applied FEM model is very simple. It is a one-dimensional model with only 16 or 25 nodes. Does the precision of system modeling require the use of a more complex model (3D, more nodes)? Must the numer of nodes increase due to a modeling error and value of misalignment. Have the tests been performed.
  2. A simple time-harmonic numerical model of the system indicates that the discussed system is linear. Have the authors can mapped the hysteresis in the system (leeway, shake of in the system)?
  3. Whether the system remains linear in terms of deformation and misalignment. Is the use of spectral decomposition justified in the study of the system due to its structure (nonlinearities, hysteresis, connection looseness)?
  4. Elements of quantitative analysis (comparison of calculation and measurement results) are available in the figures (pages 6-13). However, it is difficult to establish the error values (relative values of errors). The summary includes vague descriptions and a qualitative analysis of the results. Can the author present, compare the results of measurements and calculations and specify the error values (e.g. in a table, in the summary)?

F. My score of implications for research and practice goes to the average, but acceptable rank. In my opinion the conclusions are too general (no quantiatative analysis).

Reviewer 2 Report

This manuscript is interesting but requires improvements before it can be considered for publication.

--improve English grammar, sentence structure, word choice, paper organization, proper use of plurals, avoid very long disjointed paragraphs, improve spacing

--avoid starting sentences with AND

--improve abstract (include some results, range and error)

--avoid lumping references...discuss each one individually and in more details

--included a figure early in the discussion so that the reader can better understand the problem

--figure 1 needs improvement

     --details for call outs, dimension, tolerances throughout

     --include a cross section

--detail materials used for bearing and components

--include an error analysis

--include more design component details

--more descriptive titles are necessary for figures and tables

--include model validation process and details

--are all variables defined?

--compare model results to experiment

--provide experiment details

--error analysis/ sensitivity analysis for all figures and data

--each figure needs a discussion in the text

--definition of terms and variables in the figures is required

--values (range) of variables associated with figures is required (i.e. b j in fig 10)

--fig 17 could also use a line drawing

--proper reference format needs to be followed

--not sure if et al in list of references is proper

Round 2

Reviewer 2 Report

This manuscript has been improved in some areas, however there are still areas that require improvement before it can be considered for publication.

 --English grammar and pper structure still needs improvement.         --improve English grammar, sentence structure, word choice, paper organization, proper use of plurals, avoid very long disjointed paragraphs, improve spacing.

 -- avoid starting sentences with AND...some corrections have even used the word AND to start sentences.

All references must be discussed individually to highlight the importance of that reference.-- avoid lumping references...discuss each one individually and in more details

discuss additional design components -- -include more design component details

 -- more descriptive titles are necessary for figures and tables

how was model validated -- include model validation process and details

Round 3

Reviewer 2 Report

The revised manuscript has been improved and most points have been clarified. It now is acceptable for publication